# Microbial Keratitis in Nepal: Predicting the Microbial Aetiology from Clinical Features

**DOI:** 10.3390/jof8020201

**Published:** 2022-02-19

**Authors:** Jeremy J. Hoffman, Reena Yadav, Sandip Das Sanyam, Pankaj Chaudhary, Abhishek Roshan, Sanjay Kumar Singh, Simon Arunga, Victor H. Hu, David Macleod, Astrid Leck, Matthew J. Burton

**Affiliations:** 1International Centre for Eye Health, London School of Hygiene and Tropical Medicine, London WC1E 7HT, UK; simon.arunga@lshtm.ac.uk (S.A.); victor.hu@lshtm.ac.uk (V.H.H.); david.macleod@lshtm.ac.uk (D.M.); astrid.leck@lshtm.ac.uk (A.L.); matthew.burton@lshtm.ac.uk (M.J.B.); 2Sagarmatha Choudhary Eye Hospital, Lahan 56502, Nepal; reenapink@gmail.com (R.Y.); dassandiip@gmail.com (S.D.S.); pankajchy1987@gmail.com (P.C.); abhishek.roshan@erec-p.org (A.R.); scehdrsanjay.singh@erec-p.org (S.K.S.); 3Department of Ophthalmology, Mbarara University of Science and Technology, Mbarara P.O. Box 1410, Uganda; 4MRC International Statistics & Epidemiology Group, London School of Hygiene and Tropical Medicine, London WC1E 7HT, UK; 5National Institute for Health Research Biomedical Research Centre for Ophthalmology at Moorfields Eye Hospital NHS Foundation Trust and UCL Institute of Ophthalmology, London EC1V 9EL, UK

**Keywords:** microbial keratitis, fungal keratitis, dematiaceous fungi, clinical diagnosis, microbiology, Nepal, *Curvularia* spp., *Fusarium* spp.

## Abstract

Fungal corneal infection (keratitis) is a common clinical problem in South Asia. However, it is often challenging to distinguish this from other aetiologies, such as bacteria or acanthamoeba. In this prospective study, we investigated clinical and epidemiological features that can predict the microbial aetiology of microbial keratitis in Nepal. We recruited patients presenting with keratitis to a tertiary eye hospital in lowland eastern Nepal between June 2019 and November 2020. A structured assessment, including demographics, history, and clinical signs, was carried out. The aetiology was investigated with in vivo confocal microscopy and corneal scrape for microscopy and culture. A predictor score was developed using odds ratios calculated to predict aetiology from features. A fungal cause was identified in 482/642 (75.1%) of cases, which increased to 532/642 (82.9%) when including mixed infections. Unusually, dematiaceous fungi accounted for half of the culture-positive cases (50.6%). Serrated infiltrate margins, patent nasolacrimal duct, raised corneal slough, and organic trauma were independently associated with fungal keratitis (*p* < 0.01). These four features were combined in a predictor score. The probability of fungal keratitis was 30.1% if one feature was present, increasing to 96.3% if all four were present. Whilst microbiological diagnosis is the “gold standard” to determine the aetiology of an infection, certain clinical signs can help direct the clinician to find a presumptive infectious cause, allowing appropriate treatment to be started without delay. Additionally, this study identified dematiaceous fungi, specifically *Curvularia* spp., as the main causative agent for fungal keratitis in this region. This novel finding warrants further research to understand potential implications and any trends over time.

## 1. Introduction

Microbial keratitis (MK) is responsible for over 2 million cases of monocular blindness annually in Africa and Asia [1]. It can result in significant morbidity, including stigma and pain. Recently, there have been calls for MK to be recognised as a neglected tropical disease by the World Health Organization [2]. Causative organisms include bacteria, fungi, protozoa, and viruses. In tropical regions, fungal infections may account for more than half of reported cases, so a key distinction for management is whether the cause is bacterial or fungal [3]. Effective treatment relies on accurately and promptly diagnosing the organism responsible, usually through smear microscopy and confirmation by culture. However, anecdotally many eye care practitioners in low- and middle-income countries (LMICs) do not have access to microbiology, relying on clinical signs alone to guide treatment [4,5]. Empirical treatment is often used, but typically this is antibiotic monotherapy, as antifungals are expensive and frequently in short supply, leading to treatment delay for fungal keratitis patients [6]. Microbial keratitis typically presents with pain, conjunctival hyperaemia (redness), corneal stromal infiltration, and epithelial ulceration. Unfortunately, differentiating between bacterial, fungal, and other types of infection clinically is challenging.

Several studies have described clinical features that are more likely to be associated with fungal versus bacterial keratitis. Earlier studies have suggested that fungal keratitis (FK) might be associated with features such as Descemet’s membrane folds, serrated margins, elevated surfaces, hypopyon, and satellite lesions; however, these were limited in the authors’ sample sizes [7,8]. Subsequent, larger cross-sectional studies have reported the frequency of clinical features for bacterial keratitis (BK) and FK, rather than attempting to objectively quantify the predictive value of each sign to make an accurate diagnosis [9,10]. One study by Thomas et al. developed a scoring tool to aid in the diagnosis of FK in the absence of laboratory tests [4]. However, this tool did not incorporate other potentially useful relevant factors, including the patient’s history. Furthermore, it did not attempt to distinguish between the major fungal genera based on clinical signs. No subsequent scoring tools have been published since.

In this study, we report the aetiology of MK from a large prospective cross-sectional study in Nepal in relation to the patient presentation and clinical signs. The incidence of MK in Nepal is amongst the highest reported in the world, at 799/100,000/year [11]. Using this dataset, we explored which clinical signs are predictive of FK or BK, formulating a clinical score that can be used to make a predictive diagnosis in the absence of further investigations. We also attempted to further distinguish three clinically distinct mycological groups of FK based on clinical signs: *Fusarium* spp., *Aspergillus* spp., and infection caused by the most commonly identified ocular dematiaceous fungal pathogens (predominantly *Curvularia* spp.).

## 2. Materials and Methods

### 2.1. Ethical Statement

This study followed the tenets of the Declaration of Helsinki. It was approved by the London School of Hygiene and Tropical Medicine Ethics Committee (Ref. 14841) and the Nepal Health Research Council Ethical Review Board (Ref. 1937). Written informed consent was obtained in Nepali before enrolment. If the patient was unable to read, the information was read to them and they were asked to indicate their consent by the application of their thumbprint, which was independently witnessed. 

### 2.2. Study Design and Setting

We prospectively recruited patients at Sagarmatha Choudhary Eye Hospital (SCEH) in Lahan, Nepal, between 3 June 2019 and 9 November 2020 for a cross-sectional analysis. This formed part of the triaging assessment used to enrol eligible patients with FK into a randomised controlled trial comparing natamycin 5% to chlorhexidine 0.2%. The full protocol for this study has been published in [12]. SCEH is a tertiary ophthalmic referral hospital in southeastern Nepal that serves a population of approximately 5 million people. 

### 2.3. Study Participants

Eligible patients were adults (>18 years) with acute MK, defined as having corneal epithelial ulceration > 1 mm in diameter, corneal stromal infiltrate, and signs of acute inflammation (conjunctival hyperaemia, anterior chamber inflammatory cells, or hypopyon).

### 2.4. Clinical Findings

Demographic details and ophthalmic clinical history were collected using a structured case record form. This included the duration of symptoms, any preceding trauma, medication (conventional or traditional), and past medical and ophthalmic history. Baseline clinical assessment included visual acuity (best spectacle-corrected visual acuity, BSCVA; presenting and pinhole visual acuity), slit-lamp examination using a structured protocol including eyelid assessment, corneal ulcer features, anterior chamber (flare, cells, hypopyon shape, and size), and perforation status. The infiltrate and epithelial defect size was calculated as the mean of the maximum diameter of the infiltrate and the widest perpendicular diameter [13]. High-resolution digital photographs with and without fluorescein staining were captured.

### 2.5. Investigations

In vivo confocal microscopy (IVCM) was performed, prior to corneal sample collection, by experienced operators using the HRT II/RCM confocal microscope (Heidelberg Engineering, Dossenheim, Germany) with a previously described technique [14,15]. All the images were reviewed during the procedure in real-time and classified by type of keratitis by one experienced observer. IVCM was used in this study to detect either fungal or amoebic keratitis; the unequivocal presence of fungal hyphae or cysts on IVCM was considered diagnostic. 

Laboratory diagnosis was determined using microscopy and culture. Corneal scrape specimens were collected from the base and edge of the ulcer using a slit lamp and 21G needles after the application of topical proxymetacaine. Samples underwent processing for Gram, potassium hydroxide, calcofluor white, and lactophenol blue preparations as well as direct inoculation on solid culture media (fresh blood agar, chocolate agar, and Sabouraud dextrose agar). Media were incubated and read daily at 35–37 °C for up to 7 days for bacteria and at 25–27 °C for up to 21 days for fungi. Organism identification was performed using standard microbiological techniques.

We followed a previously described approach for reporting positive microbiological results [4]. In brief, culture results were significant if one of the following conditions were met:growth of the same organism was demonstrated on two or more solid culture media;semi-confluent growth at the site of inoculation or growth on one solid medium consistent with microscopy;semi-confluent growth at the site of inoculation on one solid medium (if bacteria);growth of the same organism on repeated scraping.

Culture positivity is the “gold standard” for the diagnosis of BK. As such, microscopy alone was not considered to be conclusive evidence if only a few organisms were seen; the exception to this rule was if many bacteria were observed in multiple fields of view. However, if fungal hyphae were visible by microscopy, the causative organism was reported as fungal (regardless of the culture results).

An overall “composite” diagnosis of definite fungal, bacterial, and mixed fungal–bacterial keratitis, or unknown aetiology, was obtained by combining the results of IVCM with cases meeting the microbiological diagnostic criteria described above. 

### 2.6. Statistical Analysis

Data were analysed in STATA 17 (STATA Corp., College Station, TX, USA). Only patients with confirmed bacterial or fungal keratitis were included in the analysis to determine the diagnostic scoring. Mixed bacterial–fungal infections were included, with a sensitivity analysis performed for non-mixed infections. Unknown cases were coded as neither bacterial nor fungal. Summary frequency tables were generated to describe the demographics, presentation time, clinical history. and features. We classified presentation time as prompt (0–3 days), early (4–7 days), intermediate (8–14 days), late (15–30 days), and very late (more than 30 days), as previously reported [16,17]. LogMAR BSCVA measurements were converted to their Snellen equivalent and categorised according to the WHO classification system [18]. Pairwise associations between clinical features (including clinical history and signs) were investigated using univariable logistic regression. Factors with univariable associations with a *p*-value < 0.2 or odds ratios (OR) greater than 2 or less than 0.5 were included in an initial multivariable logistic regression model; then, factors with associations with a *p*-value > 0.05 were removed one by one using backwards elimination. A predictive score was derived from a count of the features independently and positively associated with fungal aetiology, similar to in previous work [4]. Diagnostic accuracy indices were calculated for each score value for diagnosing FK. The probability of fungal infection was calculated by running our logistic regression model with score as the exposure, calculating the log odds of each person (based on their score) and the standard error, and converting these to probability. The 95% confidence intervals were calculated in a similar fashion. Rainfall data were obtained for Janakpur (the capital of Province 2, 56 km from Lahan) from the Department of Hydrology and Meteorology [19].

## 3. Results

### 3.1. Participants

Between 3 June 2019 and 9 November 2020, 890 patients with suspected MK were assessed at SCEH. Of these, 643 participants consented and were included in this study. The reasons for exclusion are listed in Appendix A. One patient fainted following visual acuity assessment; some clinical data are therefore missing for this patient. All cases of keratitis were unilateral (331/643, 51.5% left eye). Recruitment was paused on 24 March 2020 and resumed on 13 June 2020 due to emergency COVID-19 legislation. Demographic characteristics are presented in Table 1. The median age was 45.9 years (IQR 35.7–57.7, total range 18.1–100.1). The majority of patients were female (61.0%), Nepali (374/643, 58.2%), agricultural labourers (332/643, 51.6%), illiterate (499/643, 77.6%), and had no formal education (494/643, 76.8%). 

### 3.2. Presentation

The number of patients with MK attending SCEH varied on a monthly basis (Figure 1), with the highest numbers presenting in November and December 2019. The overall patient numbers from March 2020 were low, coinciding with COVID-19-related restrictions. Patient numbers were at their highest during the dry, winter months (October–January), which correspond to the main harvest months in the region. There was no apparent direct relationship between case numbers and the monsoon rains that occur between June and September. The median time from the onset of symptoms to presentation at SCEH was 8 days (IQR 4–13, total range 0–92 days, Table 1). Only 14% of patients presented “promptly” within 3 days of symptom onset. A definite history of trauma was reported in 49.3% (317/643) of cases. Of the cases with a history of trauma, 71.3% (226/317) reported trauma with vegetative material. Preceding use of traditional eye medication (TEM) was reported very infrequently (1.6%, 10/643), whilst 7.8% (50/643) had used topical steroids prior to attendance. Of note, 463/643 (72%) of cases reported the use of topical antibiotics prior to presentation.

### 3.3. Clinical Features and Diagnosis

Table 2 shows the clinical features at presentation. Over one-quarter of patients (26.6%) were classed as blind in the affected eye at presentation with a BSCVA of less than 3/60. The median infiltrate size and epithelial defect sizes were 2.75 mm (IQR 1.75–4.0) and 2.90 mm (2.0–4.25), respectively. 

### 3.4. Aetiology and Diagnosis of Microbial Keratitis

Combining the microbiology and IVCM results, fungi alone were identified as the main causative agent of infection, being responsible for 75.1% of MK cases (Table 3). Mixed fungal–bacterial infections were present in 50/642 (7.8%) of cases, whilst bacteria alone were identified in 33/642 (5.1%) of cases. No causative agent could be identified in 77/642 (12.0%) of cases by either IVCM or microbiology. In each case of mixed infection, a single bacterial species was associated with a single fungal species. No cases of *Acanthamoeba* keratitis were identified in this study.

There were 111/642 (17.3%) cases with no microbiological diagnosis. Microscopy and culture were negative in 84/111 cases, microscopy was positive for bacteria but cultures were negative in 17/111 cases, cultures were not performed for 6/111 cases, and microscopy was negative with no cultures performed in 4/111 cases. Of these “negative” results, 41/111 (36.9%) showed unequivocal diagnostic evidence of fungal hyphae visible by IVCM. Of the cases with bacteria detected on microscopy but no culture results, 15/23 (65.2%) were Gram-positive cocci. Bacterial infection alone was identified by microbiology in 53/642 (8.3%) of cases (i.e., no evidence of fungal infection was identified by microbiological investigations). However, 20/53 (37.7%) of these had evidence of fungal infection by IVCM and so were diagnosed as mixed fungal–bacterial infections. 

### 3.5. Fungal and Bacterial Organisms

The fungal organisms identified by culture are presented in Table 4. *Curvularia* spp. was the most frequently identified fungal genus, isolated in 170/397 (42.8%) of positive fungal cultures. Dematiaceous fungi accounted for over half of all fungal organisms cultured (201/397, 50.6%). The second and third most commonly isolated genera were *Fusarium* spp. (63/397, 15.9%) and *Aspergillus* spp. (54/397, 13.6%). It was not possible to identify the fungal organism in 51/397 (12.8%) of cases because they either failed to grow or it was not possible to induce sporulation in vitro. Two cases of yeast infection were identified (0.5%), and there were two mixed filamentous fungal infections (0.5%). 

Of the bacterial isolates identified, Gram-positive cocci (*S. aureus* (11.8%), coagulase-negative staphylococci (17.2%), and pneumococci (19.4%) were the most common cause of infection. There were just 3 cases (3.6%) of *Pseudomonas* spp. *Streptococcus* spp. was the most common bacterial genus identified (23/83, 27.7%), followed by *Staphylococcus* spp. (*Staphylococcus aureus* 6/83, 7.2%). Due to resource limitations, further identification was limited. 

### 3.6. Clinical Features and Causative Agent

The frequency of various clinical features observed in FK and BK cases (including mixed infections) is shown in Table 5. Features significantly (*p* < 0.05) associated with fungal keratitis by univariate analysis were as follows: serrated margin, absence of hypopyon, raised slough, satellite lesions, absence of nasolacrimal duct obstruction (NLDO), vegetative trauma, delayed presentation (>3 days), previous antibiotics, and previous steroids (Table 5). There was no evidence of an association between the frequency of occurrence of fibrin, reduced corneal sensation, the presence of an immune ring, keratic precipitates, perineural infiltrates, endothelial plaque, flare or cells in the anterior chamber, or previous TEM use with FK. In a multivariable logistic regression model, the clinical features predictive of fungal infection were serrated margins, the absence of NLDO, raised slough, and vegetative trauma (Table 6). The presence of NLDO, the absence of serrated margins, and no prior use of topical antibiotics were associated with BK.

A score was derived from the four clinical features associated with FK (serrated margin, raised slough, trauma with vegetative object, and absence of NLDO): a score of +1 was given for each feature present (Table 7, Figure 2). The probability of FK if only one sign was present was 30.1% (95% CI 17.8–47.2%), compared to a probability of 96.3% (95% CI 91.3–98.4%) if all four clinical features were present.

A sensitivity analysis that excluded mixed infections was carried out (Appendix A). This found that the same clinical parameters were statistically independent risk factors associated with FK. There was very little difference between the screening test indices calculated for each score or the probability of FK at different scores (Appendix A).

Univariable logistic regression found that the presence of serrated margins, raised slough, and pigmented colour, in addition to a history of preceding steroids, were risk factors for dematiaceous FK (Appendix A). The presence of satellite lesions and reduced corneal sensation were more strongly associated with other aetiologies. Only 11% of all the dematiaceous fungi isolated had pigmented corneal ulcers at presentation. The multivariate logistic regression model found that clinical features predictive of dematiaceous FK were a pigmented colour and the presence of serrated margins, as well as an absence of satellite lesions and/or fibrin (Appendix A).

No clinical parameters were found to be predictive for either *Fusarium* or *Aspergillus* keratitis by univariable or multivariable logistic regression models, possibly due to the small sample size used (data not shown).

## 4. Discussion

In this prospective study from eastern Nepal, fungal organisms were found to be the sole cause of infection in 75.1% of patients with microbial keratitis, with fungal organisms implicated in 82.9% of MK cases when mixed fungal–bacterial infections were included. To the best of our knowledge, our study reports the highest proportion of FK cases amongst MK anywhere in the world. With the highest previous reported proportion being 81.5% in Sri Lanka (1976–1981) [5,20], this is considerably higher than the proportions reported in previous studies from Nepal (ranging from 25% in Kathmandu to 70% in Biratnagar) [21,22,23,24,25,26,27].

Furthermore, another surprising finding from this study was that *Curvularia* spp. was the most frequently isolated fungal genus (42.8%). We believe that this is the only study to report a dematiaceous fungus as the leading causative organism. Over half the cases in our study were dematiaceous moulds. This is in itself unusual; FK caused by dematiaceous or melanised species is usually less common than *Fusarium* spp. and other hyaline filamentous fungi, which typically account for the majority of FK cases [28,29,30]. The only study to date with a similar proportion of dematiaceous fungi was a prospective study from North India conducted between 1999 and 2001 on 485 cases of MK, of which 39% were found to be fungal in aetiology. Although *Aspergillus* spp. (41%) was the most common fungal isolate, followed by *Curvularia* spp. (29%), dematiaceous fungi as a group (*Curvularia* spp., *Bipolaris* spp. and *Alternaria* spp.) accounted for 43.2% of cases [31]. Our finding that *Curvularia* spp. is the most commonly isolated dematiaceous genus is supported by the majority of other studies [28,32,33,34,35], with only one study from South India and one recent study from Thailand finding *Cladosporium* spp. and *Lasiodiplodia* spp. to be the most commonly isolated dematiaceous fungal genus, respectively [36,37].

Dematiaceous fungi and *Fusarium* spp. are plant pathogens which are frequently found in soil in tropical and sub-tropical regions [28,33]. As a result, one would typically expect a history of trauma, particularly with organic material, to precede FK. Indeed, a break in the corneal epithelium is required for all but a handful of microorganisms to establish an infection, which is usually the result of trauma. However, in our study only 33% of FK cases gave a definitive history of trauma, whilst 25% of BK cases also reported such a history. This proportion was similar for dematiaceous fungi alone (32%). Despite this relatively small difference, organic trauma was independently associated with FK versus non-fungal keratitis at presentation. This relatively low proportion of patients with a definite history of trauma contrasts with an earlier study from a similar location in Nepal (Nepalgunj, 2011–2012), where 58% of patients gave a history of trauma [23]. Studies from India also typically report a history of trauma greater than that found in ours, with a range of 40–92% in all studies included in a recent review of FK [5], other than one study from Delhi which had a similar number to ours (32%) [38]. This is surprising, given that the majority of the study participants were either farmers (51.5%) or likely subsistence farmers, given that they were unemployed (40.9%), as found in previous studies from Nepal and India [23,39]. The plains or “terai” area of Nepal is predominantly an agricultural society mainly involved in “paddy” farming, and it is likely that fungal spores are ubiquitous. One possible explanation for the lack of preceding trauma could be that some people may not recall minor abrasive events, which could be a sufficient breach for infections to develop.

In terms of patient demographics, the median age of 45.9 is similar to that found in other studies from the region [5]. We found there was also no change in the frequency of FK with increasing age, but that BK became slightly more frequent in patients aged 50 or older (60% of all BK cases including mixed infections compared to 42% of all non-BK cases, *p* = 0.049). BK was previously shown to be more common in older patients [36], and our results support this finding. In LMICs, MK has typically been more common in males [40], regardless of its aetiology [5]. However, in this study 61% of patients were female, with a similar female predominance found if stratified according to aetiology. There has been one other study from Nepal where the majority of cases occurred in women [25]. The reason for this is unclear, but may represent increased exposure amongst women in this region to agricultural work and trauma or possibly different health-seeking behaviours. Further epidemiological research is required to investigate this. Most patients were from low socio-economic groups, as evidenced by 76.7% reporting no formal education and 77.8% being illiterate. These findings highlight important opportunities and challenges for primary prevention strategies within Nepal.

There was considerable variation in the numbers of people presenting with MK per month; the highest numbers were seen between September 2019 and January 2020. This contrasts with the results of a five-year retrospective study conducted from the same institution between 2010 and 2015, which found little variation and with the peak attendance occurring between June and August [41]. Our observations obtained from March 2020 onwards are greatly affected by travel restrictions imposed to control the COVID-19 pandemic. The winter months in Nepal (October–January), when the numbers presenting in our study were at their highest, are the cool, dry harvest months, when the number of fungal spores in the atmosphere are likely to be at their highest [42]. Interestingly, we did not find any increase in cases during the monsoon rains. In particular, the number of dematiaceous fungal cases presenting to SCEH appear to follow this trend. It is likely that the seasonal distribution reflects periods of increased risk of trauma through occupational risk factors, such as corneal abrasions from the direct inoculation of plant material and dust during harvesting, threshing, and winnowing, which are seasonal activities intrinsically linked with climate. A recent study from North India reported a similar seasonal trend for dematiaceous fungi, with the majority of cases presenting between September and December [33]. Although we did not collect data on the number of non-microbial keratitis patients attending SCEH during this period, historically the number of cases attending is constant throughout the year, other than a slight increase in elective surgical procedures such as cataract surgery occurring between December and March. It is therefore unlikely that this observed increase is simply due to an inflated denominator.

Clinically differentiating FK from BK is challenging. The sensitivity of experienced ophthalmologists clinically diagnosing FK has been reported as very low (38%) [43], whilst corneal specialists have been shown to only be able to correctly diagnose fungal keratitis from clinical photographs in 66% of cases [44]. Several clinical signs have been found to be helpful in discriminating FK from BK: serrated margin, raised slough, colouration other than yellow, and the absence of fibrin [4]. In an earlier analysis, in patients who had raised slough, serrated margins, and no anterior chamber fibrin the probability of fungal keratitis was 89% [45], compared to 16% if the margins were defined with a flat surface and fibrin present. In our study, we also found serrated margins and raised slough to be independently predictive of FK, with higher odds ratios (95% CI in brackets) than those seen in the previous study (serrated margins OR 7.50 [4.09–13.78] vs. 3.45 [2.12–5.64]; raised slough OR 4.25 [2.51–7.24] vs. 2.32 [1.43–3.74]). We did not find colour or the absence of fibrin to be independently associated with FK. However, a history of organic trauma (OR 2.65 [1.32–5.32]) was found to be associated with fungal infection, whilst nasolacrimal duct obstruction was not (OR 0.18 [0.07–0.42]). The previous study by Thomas and co-workers only included clinical signs and therefore did not assess a positive history of trauma, as they were concerned about recall bias and the fact that, pathologically speaking, nearly all microbiological organisms require a defect in the corneal epithelium in order to enter, which is usually a result of mechanical trauma [4]. The probability of FK in our series was 96.3% if all four clinical features were present. The one clinical sign most likely to distinguish FK from BK is the presence of a serrated or irregular margin, as this was the only clinical sign whose presence or absence was found to be independently significantly associated with fungal and not with bacterial keratitis, respectively. 

Satellite lesions, which have previously been thought to be indicative of FK based on limited case reports, were not found to be significant predictors in our multivariate model, despite occurring more frequently in fungal than in bacterial cases. This is in keeping with the results from other cross-sectional studies [4,46].

Although NLDO has been suggested as a major risk factor for non-healing bacterial keratitis (BK) [47], until recently few studies have confirmed an association between NLDO and BK [48,49]. Recent work from India adds weight to this by demonstrating that patients with untreated NLDO and MK have a worse clinical outcome [50]. For this reason, at our institution all patients with MK undergo lacrimal syringing. In this study, 16% of BK patients had NLDO, compared to only 3.1% of FK patients (*p* < 0.001). We would therefore recommend clinicians consider lacrimal syringing as part of their MK diagnostic work-up, as this can help to differentiate BK from FK, as well as potentially detecting patients at risk of recurrence and a poor outcome.

Although microscopy and culture remain the “gold standard” in terms of diagnosing MK, a clinical score based on predictive factors for FK can help guide the clinician to start antifungal treatment promptly if these investigations are not possible. This also allows for the more rational use of antifungals, helping to reduce the cost to the patient and the risk of resistance whilst ensuring that the limited supplies of antifungals reach those most in need.

We found that raised slough, pigmentation, the absence of satellite lesions, and/or the absence of fibrin were clinical features predictive of dematiaceous fungal infection. A previous study from India also found that raised slough and pigmentation were predictive features for dematiaceous fungal infection [51], although it did not mention the other two clinical features. Consistent with our study, only 16% of dematiaceous cases were pigmented, whilst recent studies from India and Thailand found 18% and 26% of dematiaceous cases to be pigmented, respectively [33,37]. We did not find any features that were predictive of *Fusarium* or *Aspergillus* keratitis, likely due to the relatively small sample size. Given the delay between corneal scrape and culture results, using these four clinical signs may be helpful to guide preliminary diagnosis and management, although if microscopy is available with a quick turn-around time, this should remain the gold standard for diagnosing fungal keratitis.

Our study has several strengths. It was a large, prospective study that utilised IVCM to help identify cases of FK which may otherwise have been missed. We had a relatively low culture-negative rate. However, there were several possible limitations. Firstly, this series may not be fully representative of all MK occurring in this population, as it was conducted at a tertiary referral centre; the cases presenting to the hospital may be more severe if milder diseases can be managed effectively in the community. Furthermore, it is possible that BK is more adequately treated than FK in primary care settings, leading to fewer BK cases being referred for treatment. This possibility is supported by the widespread prior use of topical antibiotics amongst our participants (72%), whilst only 21% had used topical antifungals. The relatively low number of patients with BK in the analysed population reduces the ability of our method to adequately detect differences between groups. Secondly, we included mixed infections in our analyses in order to include as many bacterial cases as possible, in contrast to previous work [4]. This was decided at the outset in the analytical plan and was deemed a pragmatic, real-world approach. However, sensitivity analyses did not find any significant difference if mixed infections were excluded (Appendix A). Thirdly, given that only conventional diagnostic resources were available (i.e., no molecular diagnostic testing was available), it was not possible to speciate some of the pathogens at our centre. This is true of many hospital laboratories in LMICs where molecular methods are unavailable. Many genera are difficult to speciate based on phenotypic characteristics alone due to inter- and intra-species variation. Fourthly, the clinical score that we calculated to aid in diagnosis needs to be replicated and assessed in other settings where the prevalence of fungal keratitis differs. Finally, we did not detect any cases of *Acanthamoeba* in this study. There were no cases which appeared clinically suspicious for amoebic keratitis. Although we used IVCM, accurately diagnosing *Acanthamoeba* infection with IVCM can be challenging and requires a highly skilled operator [52]. No *Acanthamoeba* cysts were visualised using microscopy and culture was not performed routinely due to the very low incidence of cases reported in the region.

## 5. Conclusions

In conclusion, this large, prospective study found dematiaceous fungi to be the most common cause of FK cases in eastern Nepal (there is variation within the country). To the best of our knowledge, this study reports the highest ever proportion of FK found amongst MK cases. Although there is significant clinical variation in presentation, certain clinical signs can help to distinguish FK from other causes: specifically serrated margins, raised slough, no NLDO, and organic trauma. We also identified raised slough, pigmentation, the absence of satellite lesions, and the absence of fibrin to be predictive of dematiaceous FK. Although microscopy and culture remain the gold standard for diagnosis, using these clinical signs may help direct clinicians without access to a microbiology service, or, in cases where microscopy and culture results are negative, to a presumptive aetiology, allowing appropriate treatment to be started without delay.

## Figures and Tables

**Figure 1 jof-08-00201-f001:**
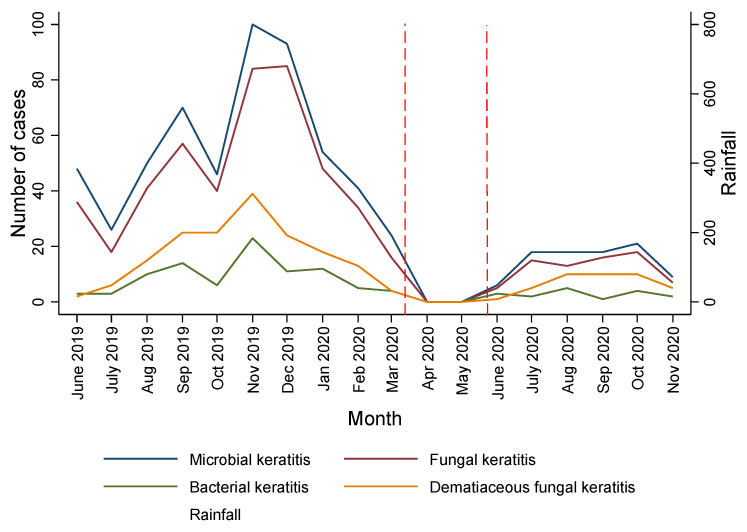
Number of microbial keratitis cases presenting per month and monthly rainfall within Province 2. Area between red dashed line represents when hospital was closed due to COVID-19 restrictions. Fungal and bacterial keratitis cases include mixed fungal–bacterial infections.

**Figure 2 jof-08-00201-f002:**
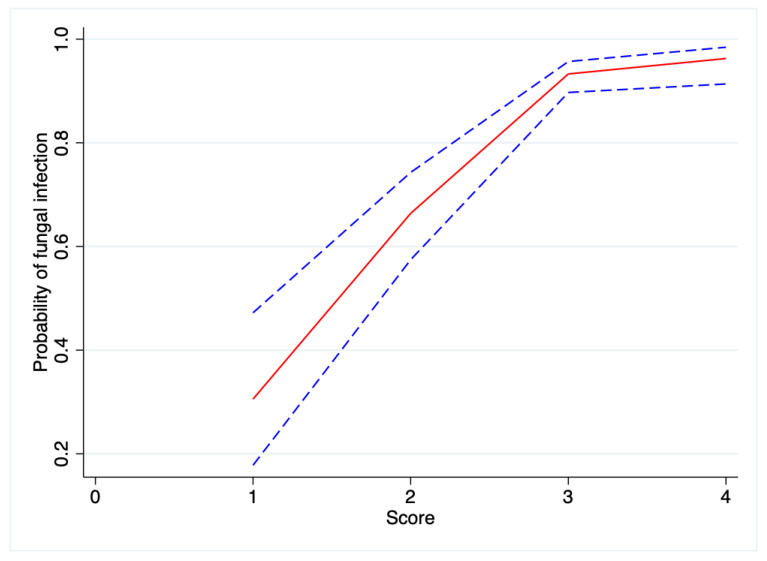
Operating characteristic curve showing the probability of fungal infection at different scores (95% CI dashed lines).

**Table 1 jof-08-00201-t001:** Demographic characteristics and clinical history of study participants.

		n/643	Percent
Age (median = 45.9, IQR 35.7–57.7)	<30 years	80	12.4%
	30–40 years	136	21.2%
	40–50 years	139	21.6%
	50–60 years	144	22.4%
	>60 years	144	22.4%
Gender	Male	251	39.0%
	Female	392	61.0%
Nationality	Nepali	374	58.2%
	Indian	269	41.8%
Occupation	No job	263	40.9%
	Farmer	332	51.6%
	Other	48	7.5%
Education	None	494	76.8%
	Primary level	80	12.4%
	Secondary level	12	1.9%
	Tertiary level	57	8.9%
Literacy level	Illiterate	500	77.8%
	Reads/writes limited Nepali	51	7.9%
	Reads/writes Nepali well	48	7.5%
	Reads/writes English and Nepali	44	6.8%
Marital status	Unmarried	66	10.3%
	Married	577	89.7%
Presenting time (median = 8, IQR = 4–13)	Prompt 0–3 days	90	14.0%
	Early 4–7 days	230	35.8%
	Intermediate 8–14 days	178	27.7%
	Late 15–30 days	108	16.8%
	Very late > 30 days	37	5.8%
Most important symptom (self-reported)	Pain	471	73.3%
	Vision	57	8.9%
	Other	115	17.9%
History of trauma	No history of trauma/unsure	326	50.7%
	Vegetative matter	226	35.1%
	Other	86	13.4%
	Unknown object	5	0.8%
Used treatment	No	93	14.5%
	Yes	550	85.5%
	Previous steroids	105	16.3%
	Previous antibiotics	463	72.0%
	Previous antifungals	134	20.8%
	Previous other topical medication	260	40.4%
	Previous systemic medication	353	54.9%
	Used traditional eye medicine	12	1.9%
Diabetic	No	630	98.0%
	Yes	13	2.0%
HIV-positive	No	643	100.0%

**Table 2 jof-08-00201-t002:** Clinical features and diagnosis at presentation.

		Median	IQR (Total Range)
Epithelial defect size (mm)		2.90	2.0–4.25 (0–12)
Infiltrate size (mm)		2.75	1.75–4.0 (0.2–11.75)
		**n/642**	**Percent**
Snellen BSCVA (affected eye) ^~^	6/5–6/18	296	46.0%
	6/24–6/60	164	25.5%
	5/60–1/60	103	16.0%
	CF-PL	80	12.4%
Slough	None	43	6.7%
	Flat	114	17.8%
	Raised	485	75.5%
Infiltrate edge	Defined	75	11.7%
	Serrated	554	86.3%
	Not visible	13	2.0%
Satellite lesions present	No	369	57.5%
	Yes	214	33.3%
	Unable to see	59	9.2%
Infiltrate colour	White	607	94.5%
	Cream	3	0.5%
	Yellow	1	0.2%
	Dark brown	10	1.6%
	Black	13	2.0%
	Other	8	1.2%
Fibrin	No	533	83.0%
	Yes	41	6.4%
	Unable to see	68	10.6%
Hypopyon	No	457	72.3%
	Yes	175	27.7%
	Unable to see	10	1.6%
Perforation status	No	634	98.8%
	Descemetocele	6	0.9%
	Perforated	2	0.3%

~ One patient fainted following visual acuity measurement; N = 643 for visual acuity but N = 642 for all other clinical features. BSCVA, best spectacle-corrected visual acuity; CF, counting fingers; PL, perception of light.

**Table 3 jof-08-00201-t003:** Aetiology of microbial keratitis with corresponding results of investigations.

	Combined Laboratory and IVCM Diagnosis (N = 642)									
	Fungal *n* (%)		Bacterial ~ *n* (%)		Mixed *n* (%)		Unknown *n* (%)		Total *n* (%)	
**Microbiological Diagnosis**										
No growth/NSS/No sample ^	41	(8.5)	0	(0)	0	(0)	70	(90.9)	111	(17.3)
Fungal keratitis	437	(90.7)	0	(0)	0	(0)	0	(0)	437	(68.1)
Bacterial keratitis	0	(0)	33	(100)	20	(40.0)	0	(0)	53	(8.3)
Mixed bacterial / fungal	0	(0)	0	(0)	30	(60)	0	(0)	30	(4.7)
Corneal scrapes not performed	4	(0.8)	0	(0)	0	(0)	7	(9.1)	11	(1.7)
**IVCM Diagnosis**										
No FK	50	(10.4)	33	(100)	6	(12.0)	77	(100)	166	(25.9)
FK	432	(89.6)	0	(0)	44	(88.0)	0	(0)	476	(74.1)
**Overall composite diagnosis (prevalence) ^#^**	482	(75.1)	33	(5.1)	50	(7.8)	77	(12.0)	642	(100)
Mixed fungal–bacterial infections included ^§^	532	(82.9)	83	(12.9)	n/a	n/a	77	(12.0)	n/a	n/a
**Results of microbiology investigations**										
Microscopy and culture-negative	34	(7.1)	0	(0)	0	(0)	50	(64.9)	84	(13.1)
Microscopy-positive, culture-negative	78	(16.2)	0	(0)	0	(0)	16	(20.8)	94	(14.6)
Microscopy-negative, culture-positive	5	(1.0)	4	(12.1)	0	(0)	0	(0)	9	(1.4)
Microscopy and culture-positive	349	(72.4)	29	(87.9)	50	(100)	0	(0)	428	(66.7)
Microscopy-positive, cultures not performed	12	(2.5)	0	(0)	0	(0)	0	(0)	12	(1.9)
Microscopy-negative, cultures not performed	0	(0)	0	(0)	0	(0)	4	(5.2)	4	(0.6)
Corneal scrape contraindicated	4	(0.8)	0	(0)	0	(0)	7	(9.1)	11	(1.7)
Total	482	(100)	33	(100)	50	(100)	77	(100)	642	(100)

^ Microscopy and culture-negative infections seen in 84/111 cases, microscopy was positive for bacteria but not meeting diagnostic criteria as no growth was seen on culture in 17/111 cases, microscopy was positive for bacteria but cultures were not performed in 6/111 cases, or microscopy was negative with no cultures performed in 4/111 cases. A total of 41/111 cases were confirmed as fungal keratitis by IVCM. ~ Bacterial keratitis was only diagnosed by significant growth on culture media, as described in the Methods Section. ^#^ Composite diagnosis was based on positive microbiological diagnosis and/or positive IVCM diagnosis. ^§^ Mixed bacterial–fungal infections (*n* = 50) were added to both fungal and bacterial categories. NSS, nothing significant seen; IVCM, in vivo confocal microscopy; FK, fungal keratitis.

**Table 4 jof-08-00201-t004:** Identification of fungi isolated from corneal samples of patients with microbial keratitis.

Fungi	*n*	Percent
*Fusarium* spp.	63	15.9
*Aspergillus* spp.	54	13.6
Dematiaceous fungi	201	50.6
*Curvularia* spp.	(170)	(42.8)
*Bipolaris* spp.	(19)	(4.8)
*Exserohilum* spp.	(7)	(1.8)
*Alternaria* spp.	(5)	(1.3)
*Scedosporium apiospermum*	2	0.5
*Sarocladium* spp./*Acremonium* spp.	8	2.0
*Pestalotiopsis* sp.	1	0.3
*Colletotrichum* spp.	6	1.5
*Purpureocillium lilacinum*	2	0.5
*Trichoderma* spp.	3	0.8
*Syncephalastrum racemosum*	1	0.3
*Fusarium* sp. and *Bipolaris* sp.	1	0.3
Mixed FFI	2	0.5
Yeast	2	0.5
Unidentified fungus	51	12.8
**Total**	**397**	**100.0**

FFI: filamentous fungal infection.

**Table 5 jof-08-00201-t005:** Clinical features occurring in fungal and non-fungal keratitis (mixed infections included), with univariable analysis for features associated with fungal keratitis.

					Indices for Detecting Fungal Keratitis						
	Frequency in Fungal Cases (Including Mixed)	(%)	Frequency in Non-Fungal Cases	(%)	Odds Ratio for FK	*p*-Value	95% CI	Sens.	Spec.	PPV	NPV
Serrated margins	497/527	94%	57/102	56%	13.08	<0.001	7.64–22.38	94.3%	44.1%	89.7%	60.0%
Fibrin	35/481	7.3%	6/93	6.5%	1.14	0.777	0.46–2.79	7.3%	93.5%	85.4%	16.3%
Hypopyon	136/524	26%	39/108	36%	0.62	0.033	0.40–0.96	26.0%	63.9%	77.7%	17.1%
Raised slough	439/532	83%	46/110	42%	6.57	<0.001	4.23–10.20	82.5%	58.2%	90.5%	40.8%
Satellite lesions	192/483	40%	22/100	22%	2.34	0.001	1.41–3.88	39.8%	78.0%	89.7%	21.1%
Pigmented colour	28/532	5.3%	3/110	2.7%	1.98	0.268	0.59–6.64	5.3%	97.3%	90.3%	17.5%
Nasolacrimal duct obstruction	15/486	3.1%	18/99	18%	0.14	<0.001	0.07–0.30	3.1%	81.8%	45.5%	14.7%
Reduced corneal sensation	70/532	13%	20/110	18%	0.68	0.169	0.40–1.18	13.2%	81.8%	77.8%	16.3%
Trauma with vegetative object	177/532	33%	21/110	19%	2.11	0.004	1.27–3.51	33.3%	80.9%	89.4%	20.0%
Previous antibiotics	392/532	74%	70/110	64%	1.60	0.034	1.04–2.47	73.7%	36.4%	84.8%	22.2%
Delayed presentation > 3 days	464/532	87%	88/110	80%	1.71	0.049	1.00–2.90	87.2%	20.0%	84.1%	24.4%
Previous steroids	96/532	18%	9/110	8.2%	2.47	0.013	1.20–5.06	18.0%	91.8%	91.4%	18.8%

FK, fungal keratitis; CI, confidence interval; Sens., sensitivity; Spec., specificity; PPV, positive predictive value; NPV, negative predictive value.

**Table 6 jof-08-00201-t006:** Multivariable analysis of clinical features occurring in fungal and bacterial keratitis (mixed infections included).

	Odds Ratio	*p*-Value	95% CI
**Fungal keratitis—clinical features**			
Serrated margins	7.50	<0.001	4.09–13.78
Raised slough	4.27	<0.001	2.51–7.24
Nasolacrimal duct obstruction	0.18	<0.001	0.07–0.42
Trauma with vegetative object	2.65	0.006	1.32–5.32
**Bacterial keratitis—clinical features**			
Serrated margins	0.36	0.001	0.20–0.66
Nasolacrimal duct obstruction	3.08	0.006	1.38–6.87
Previous antibiotics	0.33	<0.001	0.20–0.53

**Table 7 jof-08-00201-t007:** Screening test indices for each score.

N = 574	*n*	(%)	Sensitivity	Specificity	PPV	NPV
Score > 0	572	(99.7)	100%	2.17%	84.3%	100%
Score > 1	536	(93.4)	97.7%	29.3%	87.9%	71.1%
Score > 2	417	(72.7)	81.3%	72.8%	94%	42.7%
Score > 3	134	(23.3)	26.8%	94.6%	96.3%	19.8%

Only patients who had all features examined were included in calculating the diagnostic accuracy. PPV, positive predictive value; NPV, negative predictive value.

## Data Availability

The datasets generated during and/or analysed during the current study will be available upon request from M.J.B. (matthew.burton@lshtm.ac.uk). The full data set will be available with all patient identifiable details removed. Data will be available after formal reporting of the study findings in a peer-reviewed scientific publication. Datasets will only be available to bona fide scientific investigators. Requests should be made to the Chief Investigator in writing detailing the scientific investigators background and intended use for the data. Consideration will be given to all proposed analyses, with likely envisaged uses including investigators planning on conducting meta-analyses, for example. Patient Information Sheets and consent forms specifically referenced making anonymised data available, and this has been approved by the relevant ethic committees.

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
