# Peer review of "Microbial Keratitis in Nepal: Predicting the Microbial Aetiology from Clinical Features"

_jof, 2022, doi:10.3390/jof8020201_

Round 1

Reviewer 1 Report

The manuscript is well written and of interest to clinicians.

The file received from JOF (figure 1) does not display the rainfall.  Rainfall was not mentioned as a risk factor within the text. Because rainfall was examined, a comment about any relationship to disease would be useful. Presumably June-September were the monsoonal months?

What is the monthly distribution of patients presenting with eye complaint of any type? How do infectious etiologies compare with non-infectious etiologies by month?

Reviewer 2 Report

Dears authors 

Keratitis is a common clinical problem in South Asia. However, it is often difficult to distinguish it from other etiologies, such as bacteria or acanthamoeba. In this prospective study we investigated the clinical and epidemiological features that can predict the microbial etiology of microbial keratitis in Nepal. Although microbiological diagnosis is the "gold standard" for determining etiology, certain clinical signs can help direct the physician to a presumed infectious cause, allowing appropriate treatment to be initiated without delay. Furthermore, this study identified dematiaceous fungi, in particular Curvularia spp., As the main causative agent of fungal keratitis in this region. This new discovery justifies further research to understand the potential implications and any trends over time.
There are some points to review and others to deepen:

1 - Introduction: the contents and writing of the general part must be reformed to review the syntax of the theme

2- Discussion: to deepen in consideration of the problem of antibiotic resistance the use of new disinfectants against multidrug-resistant strains that cause keratitis, these are new ophthalmic treatment perspectives. Find out more about this by using and citing the following references: PMID: 32452982 ;  PMID: 34829196 ; PMID: 32524193  
3 - Check the bibliographic entries throughout the text, some of which are non-compliant, review some entries in the bibliographic references and necessarily insert those referred to in point 2 for the purpose of my acceptance.

4 - Review the English grammar and in particular the applied scientific English: in particular the verb tenses and the syntax in the discussion.
